# Design and Simulation of a Bio-Inspired Deployable Mechanism Achieved by Mimicking the Folding Pattern of Beetles’ Hind Wings

**DOI:** 10.3390/biomimetics10050320

**Published:** 2025-05-15

**Authors:** Hongyun Chen, Xin Li, Shujing Wang, Yan Zhao, Yu Zheng

**Affiliations:** College of Mechanical and Electrical Engineering, Suqian University, Suqian 223800, China; hongyccc@126.com (H.C.); 18119@squ.edu.cn (Y.Z.); yuzhengnuaa@163.con (Y.Z.)

**Keywords:** biomimetic deployable mechanism, beetle hind wings, folding performance, neural network, structural response

## Abstract

In this paper, a beetle with excellent flight ability and a large folding ratio of its hind wings is selected as the biomimetic design. We mimicked the geometric patterns formed during the folding process of the hind wings to construct a deployable mechanism while calculating the sector angles and dihedral angles of the origami mechanism. In the expandable structure of thick plates, hinge-like steps are added on the thick plate to effectively avoid interference motion caused by the folding of the thick plate. The kinematic characteristics of two deployable mechanisms were characterized by ADAMS 2018 software to verify the feasibility of the mechanism design. The finite element method is used to analyze the structural performance of the deployable mechanism, and its modal response is analyzed in both unfolded and folded configurations. The aerodynamic generation of a spatially deployable wing is characterized by computational fluid dynamics (CFD) to study the vortex characteristics at different frame rates. Based on the aerodynamic parameters obtained from CFD simulation, a wavelet neural network is introduced to learn and train the aerodynamic parameters.

## 1. Introduction

Micro air vehicles (MAVs) are increasingly being used in search and rescue missions due to their ability to quickly collect and transmit information from remote and dangerous areas. When natural disasters occur, MAVs are used for imaging and locating the location of victims, which helps improve the efficiency of rescue efforts [1,2,3]. Usually, the flight environment data of multi-rotor drones during mission execution is constantly changing, and they cannot hover, capture, and locate narrow spatial environments well. Meanwhile, fixed-wing aircraft can only achieve optimal aerodynamic performance at one design point, but there are technical bottlenecks in improving their flight performance and multi-mission adaptability. In this environment, fixed-wing MAVs with multi-rotor and non-foldable wings gradually cannot meet the application requirements of current multi-objective missions. In fact, the deformable wing MAV benefits from the changes in its wing geometry that can adapt to different environmental conditions [4,5,6]. With the miniaturization of deformable wing aircraft, their flight Reynolds number has significantly decreased. Therefore, deformable wing MAVs can effectively avoid collision risks when performing missions in narrow spaces due to their excellent low-speed flight and hovering capabilities.

The exploration of the flight principles of animals observed in nature can greatly improve the maneuverability of existing aircrafts and promote the development of new and unique aircrafts. Compared with traditional aircraft, MAVs that mimic insect flight and wing structures have advantages of simpler design, lower noise, higher efficiency, and better environmental protection. However, observing the folding morphology and flight mechanism of insect wings is a relatively new and complex research field. Dermaptera insects can store their flexible hind wings with high folding rates under their hard forewings and then drill into the soil [7,8,9,10,11,12]. At present, the biomimetic design of foldable mechanisms mainly considers the factors of simple structure and flexibility, which limits the improvement of folding and unfolding motion performance. Therefore, focusing on a suitable flying animal is extremely important for guiding the biomimetic design and folding deformation mechanism of flexible foldable wings.

For example, the morphological changes in bird wings are modeled by designing a deformable wing driven by centrifugal acceleration [13], which can automatically fold upon collision, reducing the weight of the driving mechanism in flapping MAVs. The folding mechanism used in flapping-wing MAVs can reduce the weight of the driving mechanism, but the folding and unfolding motions of the wings during flight are not easy to control. By imitating the feather distribution pattern on pigeon wings, a deformable wing with multiple elastic connected feathers was developed [14], which rapidly deforms under aerodynamic loads. Although the deformed wing can reduce wing damage when dropped, further improvements are needed in terms of lightweight and miniaturization. Design a biomimetic bat with controllable folding motion of wings by mimicking the multi-joint folding morphology of bat wings [15]. However, the folding mechanism has a low folding ratio and inflexible folding motion after attaching the elastic wing membranes. The fan-shaped folding pattern of the lateral and longitudinal folding ability of earwig’s hind wings has been biomimetically designed [16]. Furthermore, the folding motion of the fan-shaped folding mechanism is achieved by the motor rotating and pulling the thin rope, which may break the rope.

In recent years, the folding mechanism of beetle wings has injected new ideas into the design of transformable wing aircraft. The folding mechanism of the hind wings of the *Allomyrina dichotoma* was analyzed, and a four-panel folding wing was proposed [17], which drives the folding motion of the artificial wing by heating shape memory alloy wires. To increase the folding ratio of artificial wings, elastic joints are used to replace the intersection points of folds and demonstrate the feasibility of flapping wing motion [18]. To further conform to the stability of flapping wings, a double four-panel mechanism was used to design artificial foldable wings [19]. However, the folding ratio of the artificial wing is too small, and the connection between the wing root and ornithopter needs to be designed separately, which increases the weight of the whole ornithopter. A hydraulic-driven artificial foldable wing is constructed using a hydraulic mechanism that mimics the wing veins of the *Dorcus titanus platymelus* beetle’s hind wings [20], and its feasibility still needs to be verified by physical prototypes. The pattern of wing vein changes during the folding process of the *Buqueti bamboo* weevil’s hind wings was designed using a linkage mechanism [21]. The rigid flexible coupling folding simulation of the flapping wing MAV and the flapping performance testing of its physical prototype have verified the feasibility of the design model, but its flight maneuverability still needs further improvement. The folds on the hind wings of *Allomyrina dichotoma* beetles not only have a shock-absorbing effect but also affect the passive retraction speed of the wing roots. Therefore, flapping-wing MAVs with collision recovery at the tip of the wings [22,23] and flapping-wing MAVs [24] with passive unfolding of the wings by the wing roots were designed. However, the folding motion of their wings during flight collisions is not significant.

As of now, the availability of MAVs is facing challenges in terms of their endurance and reduced payload capacity. Miniaturization and foldability are the preferred solutions for enhancing the portability of drones and promoting the development of drone miniaturization technology. Foldable drones can be large enough to carry useful payloads when fully deployed and convenient for transportation after folding. In this paper, the *Protaetia brevitarsis* beetle with a large folding ratio of its hind wings is used to biomimetic design a spatially deployable biomimetic deformable wing MAV. Firstly, the morphological transformation patterns during the folding motion of the hind wings were observed. The folding angle of the biomimetic origami model is calculated, and a spatially deployable MAV model with a larger folding rate is constructed. Then, ADAMS and ANSYS 19.0 software were used to analyze the folding performance and structural response characteristics of spatially deployable mechanisms. Finally, a computational fluid dynamics (CFD) solver based on an in-house immersed boundary method was used to characterize the aerodynamic performance of the deformable wing MAV. To reasonably evaluate the generated aerodynamic force, the introduced neural network model was trained. The simulation of MATLAB 2021a software verified that the network model can reasonably predict the thrust and lift characteristics generated by the deformed wing MAV in the second half of the cycle.

## 2. Design and Analysis of Biomimetic Space Deployable Mechanism

### 2.1. Folding Pattern of Hind Wings

The *Protaetia brevitarsis* beetle was captured in Suqian City, Jiangsu Province, in July 2024. The wings of the dead beetle were cut from the wing roots and soaked in ethanol to observe the folding pattern of the hind wings. This beetle has good flight ability and aerodynamic characteristics [25,26,27]. As shown in Figure 1, the hindwing is mainly composed of wing veins, folding joints, and relatively soft membranes. During flight, the macroscopic morphology of the hind wings is maintained by a thick framework called the venation pattern. The wing root contains three thick main wing veins, namely the Costa (C), Radius Posterior (RP), and the MP1+2 branch veins of the Media Posterior (MP), which converge at the wing root point to form a middle bridge [28]. The driving muscles for wing folding and unfolding motions are concentrated at the wing roots, and complex-shaped skeletal muscle tissue can be observed. The wingtip of the hind wing is mainly supported by the leading edge frame, which includes two main wing veins and displays the folding pattern near the wingtip in a Z-shaped style [29].

The hind wings fold longitudinally and transversely along the wing, in spanwise and chordwise directions, and the transverse fold line causes the hind wings to undergo two folding motions and form a Z-shape. These crease lines intersect in the central area of the hind wings and create a complex crease pattern. The folding pattern of the rotational joint position on the hind wings can be simplified into a triangular folding pattern with multiple four-degree vertices, as shown in Figure 1. The foldability condition for the design of the beetle-inspired pattern is to obtain the sector angles within each vertex. Angles p_1_ and p_2_ define the pivot lines, while m_1_ and m_2_ define the median lines, and *P* and *M* are the base points of these folding lines. The three four-degree vertices are named *V*_1_, *V*_2_, *V*_3_, and *V*_4_, with sector angles *α_i_*, *β_i_*, *γ_i_*, and *δ_i_*, where *i* = 1, 2, 3, and 4 are defined at each vertex. All vertices are located on the same plane when the hind wings are fully unfolded. Therefore, the condition for the foldability of a flat panel can be expressed as follows:(1)$$\alpha_{i}+\gamma_{i}=\pi (i=1,2,3,4)$$(2)$$\beta_{i}+\delta_{i}=\pi (i=1,2,3,4)$$

Due to the four triangles surrounding *V*_1_ forming a quadrilateral with two common lines. According to the sine theorem, the following equation can be obtained [30]:(3)$$\frac{\sin p_{1}}{\sin m_{1}}\frac{\sin m_{2}}{\sin\beta_{2}}\frac{\sin\alpha_{2}}{\sin\gamma_{3}}\frac{\sin\beta_{31}}{\sin p_{2}}=1$$

When all vertices are connected and form a closed loop, an additional condition is required so that all folding angles are determined and no folding interference occurs. As shown in Figure 2, in the four-angle single-vertex model (four-panel folding model) that can achieve planar folding, the angle transformation coefficients *p* between ρ_0_ and ρ_3_ are as follows:(4)$$p(\alpha ,\beta )=\frac{1-\tan\frac{\alpha }{2}\tan\frac{\beta }{2}}{1+\tan\frac{\alpha }{2}\tan\frac{\beta }{2}}$$

For the convenience of analysis, the folding pattern of the hind wings in Figure 1 is equivalent to the unfolding process of the cardboard shown in Figure 2. The crease marking line pattern in Figure 1 is equivalent to the four-panel form in Figure 2 when it satisfies the condition of rigid foldability. The product of the angle transfer factors in the first three vertices *V*_1_→*V*_2_→*V*_3_→*V*_1_ in Figure 1 should be equal to 1. Namely [30],(5)$$\frac{p(\pi -\alpha_{1},\pi -\beta_{1})p(\pi -\gamma_{2},\pi -(\alpha_{1}-m_{2}))}{p(\pi -(\alpha_{1}+p_{2}),2\pi -(\gamma_{2}+\beta_{1}))}=1$$

In fact, real beetle hind wings can complete flexible folding motions without considering rigid folding conditions. When rigid folding plates are used to design biomimetic deployable wings, the condition of rigid foldability becomes very important. It is necessary to calculate the sector angle of each flat plate to avoid the interference problems during the folding motion of thick plates.

### 2.2. Design of Biomimetic Space Deployable Mechanism

The expected folding pattern of biomimetic foldable wings is to achieve folding motion at the maximum folding ratio and require a single degree of freedom to achieve unfolding motion. Referring to the folding pattern of the hind wings of the *Buqueti* bamboo weevil [31], the folding pattern of the hind wings of the *Protaetia brevitarsis* beetle can be equivalent to a double four-plate folding model, and the feasibility of its folding motion can be simulated and analyzed using ADAMS software. Then, based on the double four-panel folding model, a spatially deployable wing MAV composed of multiple thick plates can be designed by vertical and horizontal chain expansions. According to the four-panel folding models in Figure 1 and Figure 2, the sector angles of each cardboard need to be calculated. The homogeneous transformation matrices **Q**_12_, **Q**_23_, **Q**_14_, and **Q**_43_ between adjacent panels 1 and 2, 2 and 3, 1 and 4, and 4 and 3 are, respectively, as follows [31]:(6)$$\begin{array}{l} Q_{12}=\left[\begin{array}{cccc} c\beta & s\beta c\theta_{1} & -s\beta c\theta_{1} & 0\\ -s\beta & c\beta c\theta_{1} & -c\beta s\theta_{1} & 0\\ 0 & s\theta_{1} & c\theta_{1} & 0\\ 0 & 0 & 0 & 1 \end{array}\right],Q_{23}=\left[\begin{array}{llll} c\gamma & -s\gamma & 0 & 0\\ s\gamma c\theta_{2} & c\gamma c\theta_{2} & -s\theta_{2} & 0\\ s\gamma s\theta_{2} & c\gamma s\theta_{2} & c\theta_{2} & 0\\ 0 & 0 & 0 & 1 \end{array}\right]\\ Q_{14}=\left[\begin{array}{llll} c\alpha & s\alpha c\theta_{4} & -s\alpha s\theta_{4} & 0\\ -s\alpha & c\alpha c\theta_{4} & -c\alpha s\theta_{4} & 0\\ 0 & s\theta_{4} & c\theta_{4} & 0\\ 0 & 0 & 0 & 1 \end{array}\right],Q_{43}=\left[\begin{array}{llll} c\delta & -s\delta & 0 & 0\\ s\delta c\theta_{3} & c\delta c\theta_{3} & s\theta_{3} & 0\\ -s\delta s\theta_{3} & -c\delta s\theta_{3} & c\theta_{3} & 0\\ 0 & 0 & 0 & 1 \end{array}\right] \end{array}$$
where *sθ_i_* = sin*θ_i_* and *cθ_i_* = cos*θ_i_* (*i* = 1~4); further calculations show that the following:$$Q_{12}Q_{23}=\left[\begin{array}{cccc} c\alpha c\beta -s\alpha s\beta c\theta_{1} & c\theta_{4}\left(c\beta s\alpha +c\alpha s\beta c\theta_{1}\right)-s\beta s\theta_{1}s\theta_{4} & -s\theta_{4}\left(c\beta s\alpha +c\alpha s\beta c\theta_{1}\right)-s\beta c\theta_{4}s\theta_{1} & 0\\ -c\alpha s\beta -c\beta s\alpha c\theta_{1} & -c\theta_{4}\left(s\alpha s\beta -c\alpha c\beta c\theta_{1}\right)-c\beta s\theta_{1}s\theta_{4} & s\theta_{4}\left(s\alpha s\beta -c\alpha c\beta c\theta_{1}\right)-c\beta c\theta_{4}s\theta_{1} & 0\\ -s\alpha s\theta_{1} & c\theta_{1}s\theta_{4}+c\alpha c\theta_{4}s\theta_{1} & c\theta_{1}c\theta_{4}-c\alpha s\theta_{1}s\theta & 0\\ 0 & 0 & 0 & 1 \end{array}\right]\;Q_{14}Q_{43}=\left[\begin{array}{cccc} c\delta c\gamma -s\delta s\gamma c\theta_{3} & -c\gamma s\delta -c\delta s\gamma c\theta_{3} & -s\gamma s\theta_{3} & 0\\ s\delta \left(s\theta_{2}s\theta_{3}+c\gamma c\theta_{2}c\theta_{3}\right)+c\delta s\gamma c\theta_{2} & c\delta \left(s\theta_{2}s\theta_{3}+c\gamma c\theta_{2}c\theta_{3}\right)-s\delta s\gamma c\theta_{2} & c\gamma c\theta_{2}s\theta_{3}-c\theta_{3}s\theta_{2} & 0\\ c\delta s\gamma s\theta_{2}-s\delta \left(c\theta_{2}s\theta_{3}-c\gamma c\theta_{3}s\theta_{2}\right) & -c\delta \left(c\theta_{2}s\theta_{3}-c\gamma c\theta_{3}s\theta_{2}\right)-s\delta s\gamma s\theta_{2} & c\theta_{2}c\theta_{3}+c\gamma s\theta_{2}s\theta_{3} & 0\\ 0 & 0 & 0 & 1 \end{array}\right]$$

According to the spatial position constraint of plate 2 in Figure 2, there is $Q_{12}Q_{23}=Q_{14}Q_{43}$ and, furthermore, there is:(7)$$\begin{array}{l} \theta_{1}=\theta_{3},\\ \theta_{2}=\theta_{4}=\frac{c\gamma s\delta +c\theta_{3}s\gamma c\delta }{s\delta s\theta_{3}+\left(s\gamma c\delta +c\theta_{3}c\gamma s\delta \right)\left(s\gamma s\delta -c\theta_{3}c\gamma c\delta +c\theta_{3}\right)/\left(s\theta_{3}c\gamma -s\theta_{3}c\delta \right)} \end{array}$$

It can be seen that the folding motion between adjacent four panels follows the angular motion relationship of Formula (7). As long as the sector angles *α*, *β*, *γ*, and *δ* of each panel are determined, the sector angle values of the entire expandable panel mechanism can be calculated. Note that the biomimetic design of an equivalent double four-panel mechanism imitating the hind wings of beetles was first carried out, and the feasibility of folding motion was simulated and verified using ADAMS software. Next, the four-panel model is extended along the wing in spanwise and chordwise directions to construct a spatially deformable wing MAV. After calculation, the sector angles of the folding panel were determined (*α* = 75°, 85°, *β* = 80°, 60°, *γ* = 105°, 95°, and *δ* = 100°, 120°), and a double four-panel folding model was constructed as shown in Figure 3.

To verify the folding performance of the six panels (Ban 1-Ban 6), the ADAMS simulation process is shown in Figure 4. During the folding process, there was no mutual interference between the plates, and they were able to fold completely as expected. The centroid changes in each cardboard during the folding process are shown in Figure 5, and the cardboard completes the folding motion smoothly within 3.5 s. To analyze the structural response of the six-panel model during folding motion, its modal shapes are shown in Figure 6. The first mode of vibration is that the four corners of the thick panel undergo warping deformation. The second mode shape is the diagonal bending deformation of the thick panel in the opposite direction. The third mode shape is the bending motion of the thick panel folding mechanism at the connection of each panel. The fourth mode shape is caused by warping in the central region of the thick folding panel mechanism. At the fifth mode, a single corner undergoes warping motion. The sixth mode shape is the symmetric bending deformation of the outer edge of the folding panel mechanism. The modal frequencies of the first two orders are reasonable, and the design of the deployable mechanism is feasible, which enables the subsequent design of a deformable wing MAV composed of multiple thick panels.

According to the design method of the six-panel folding model, the sector angle of each panel is calculated, and a spatial deformation wing model composed of multiple thick plates is constructed. Based on vertex *V*_1_, *β*_1_ (116°) is predetermined, and the sector angles of each folding panel are calculated sequentially as *α*_1_ = 64°, *γ*_1_ = 89°, and *δ*_1_ = 91°. The same calculation method is used for the angles of the four panels involved in vertices *V*_2_–*V*_4_. Due to the difference between thick plate folding and origami folding, the sector angles may differ by 2–3° when designing thick panel folding wings. Unlike the method of leaving characteristic gaps in the thick panel of the physical prototype in reference [32] to deal with interference phenomena that may cause penetration during the folding of thick plates, here, a step hinge is set at the lower end of the edge of the folded thick plate with a thickness equal to the sum of the thicknesses of the other two plates. The 3D model of the spatially deployable deformable wing MAV constructed after calculating all sector angles is shown in Figure 7. The folding motion of these thick plates follows the bistable mechanism of multiple four-plate mechanisms during folding motion.

### 2.3. ADAMS Simulation of Biomimetic Space-Deformable Wing MAV

In order to verify the effectiveness of the folding motion of the spatially deployable deformable wing MAV, ADAMS software was used for simulation. In the visual interface of ADAMS/view, by defining the constraint relationships between thick panels, a motion model of a spatially deployable mechanism can be established. We applied reasonable rotational motion pairs on the hinges connecting each thick panel. The driving time function was set between two thick plates near the propeller, and the simulation time steps were set to 1500 steps. The simulation folding process is shown in Figure 8, and it can be seen that each thick panel can be folded in order without interference or penetration between the cardboard during the folding motion.

### 2.4. Modal Characteristics of Biomimetic Space-Deformable Wing MAV

Modal analysis can help to gain a preliminary understanding of the structural response of airfoil dynamic changes before CFD analysis, which has guiding significance for analyzing the aerodynamics of deformed wing MAVs. In ANSYS Workbench, each thick panel (3 mm) is made of a foam plate with a modulus of 50 MPa, Poisson’s ratio of 0.38, and density of 30 kg/m^3^. The propeller and tail wing are set as fixed constraints. As shown in Figure 9, the grid size in the unfolded configuration is set to 4 mm, and the number of nodes and elements after meshing is 79,811 and 35,899, respectively. The grid size in the folded configuration is set to 1 mm, and the number of nodes and elements after meshing is 627,360 and 278,458, respectively. The area of the entire wing surface in the unfolded configuration was measured using NX 12.0 software to be 74,553.8017 mm^2^. The design objective based on a physical prototype [32] with a weight of 26 g was applied to a pressure load of q = G/A = 0.348741 Pa on the wing surface.

## 3. Results and Discussion

### 3.1. Kinematics of the Deformable Wing MAV

The motion pattern of each cardboard (ban1-ban4) during the folding motion of the expandable mechanism formed by vertex V_4_ is shown in Figure 10. It can be seen that the folding process is relatively smooth, with no intersection points on the curves, which also indicates that there is no penetration between each panel. After 1.5 s, the curves converge at one point, indicating that the four panels are completely folded and each panel is not adhered together. The speed variation of the cardboard during the folding process is shown in Figure 11. When the folding motion occurs during the first 0.25 s, the unstable folding speed is influenced by the bistable mechanism between the four-plate mechanism. There are multiple peaks in the velocity curve, which is due to the coupling connection between the four-plate mechanism and adjacent thick panels. The sudden inertial force phenomenon caused by the folding motion of a certain thick plate leads to the sudden fluctuation of the velocity curve. During the time period of 0.25–1.5 s, the folding motion changes smoothly, and there is no vibration phenomenon. The speed of the thick panel decreases gradually until the speed of the thick panel reaches 0, at which point the folding motion is completed.

### 3.2. Structural Response of the Deformable Wing MAV

Modal analysis of a wing is conducted under free boundary conditions without external constraints to accurately simulate the behavior of an aircraft during flight. The first six modes of vibration of the deformable wing MAV are shown in Figure 12, and their natural frequencies are shown in Table 1. The first mode of vibration for the wing is pure bending mode with a natural frequency of 8.3839 Hz. The wing bends along the Z-axis in the vertical direction because it tends to bend near the minimum moment of inertia at the root section. The maximum total deformation is developed at the wingtip of 0.9188 m. Similar to the first-order pure bending mode, the second-order mode of vibration has a natural frequency of 12.0280 Hz. The maximum total deformation occurs at the right wingtip and the tip chord of the left wing, which is 0.98105 m. The mode of the third-order vibration exhibits bending and torsional deformation. The twisting from the bottom to the top of the wing occurs at the trailing edge of the wingtip chord, with a maximum total deflection of 0.12539 m at the trailing edge of the wing. From the fourth mode shape, it can be seen that another torsional motion at the tip chord of the wing has begun to develop. The only reason for this situation is that the thick plate connection cannot resist the pressure load applied by the bottom surface of the wing. For the fifth mode of vibrations, it occurs when the natural frequency is 65.045 Hz, and the wing experiences a pure torsional failure in the counterclockwise direction. The sixth mode of vibration shows torsional deformation occurring at the trailing edge and leading edge tip of the wing, with the smallest deformation occurring at the root near the trailing edge of the wing. A high-deformation local area is formed at the bottom of the trailing edge, and the radial torsional deformation response on the wing is similar to a cantilever beam with two opposite point loads. In fact, drones in flight can move without restrictions because they have three translational and three rotational degrees of freedom [33]. The modal analysis of the deformed wing MAV in a self-excited state replicates this unconstrained state to ensure that the calculated natural frequencies and modes are very similar to the performance of the wing under actual flight conditions [34]. In the optimization design of MAV, the first two modal shapes of the wing are the main focus, because the natural frequency of the wing cannot coincide with the front modal frequency under real testing [35], which can avoid the resonance phenomenon of the wing.

Similar to the modal analysis in the unfolded configuration, as shown in Figure 13. The first six natural frequencies of the deformed wing MAV in the folded configuration are significantly increased, as shown in Table 2. The first mode exhibits common bending deformation in the wing spanwise direction. The second-order mode shape is the diagonal bending of the thick panels in the opposite direction. The third-order modal shape is warped at the connection between thick panels. The fourth-order mode of vibration is characterized by warping in the middle region of the thick panel, while the fifth- and sixth-order modes are characterized by symmetrical bending at the outer edge of the thick panel. In addition, the deformed wing MAV exhibits bending deformation of both wings in the first two modes, and the bending deformation is relatively coordinated, which is consistent with the bending deformation characteristics formed by the aerodynamic load during MAV flight. From the third mode to the sixth mode, the wing of the deformed wing exhibits bending and torsional deformation. The higher the modal frequency, the more pronounced the wing vibration and the greater the torsional deformation, which increases the risk of wing fracture [36].

To investigate the structural performance of deformable wing MAVs under pressure loads, the load-bearing stability of the wing needs to be analyzed. The area of the entire wing surface in the unfolded configuration was measured using NX 12.0 software to be 74,553.8017 mm^2^. Based on the design goal of a physical prototype with a weight of 26 g [30], the pressure load applied to the wing surface was q = G/A = 0.348741 Pa. When the surface is subjected to a vertical upward pressure load, the directional deformation and total deformation of the wing can be obtained, and the entire wing surface must undergo gradual deformation as it moves away from the fixed end. In the total deformation shown in Figure 14d, the maximum deformation of the wing occurs at the wingtip, with a value of 0.32219 mm, which can be ignored compared to the wingspan of the wing. The total deformation of the wing exhibits good deformation coordination and is also analyzed in cambered and corrugated wings [26,37,38,39], which is an important consideration in the structural design of the wing. As shown in Figure 14b,c, the maximum deformation of the wing along the Y-axis and Z-axis directions is more pronounced after the wing surface is loaded. The maximum deformation along the X-axis direction is mainly manifested as normal bending deformation after being loaded, which radiates from the central region to the wingtip. This is because the hinge step is the main area responsible for the repeated folding and unfolding movements of the mechanism. The deformation of the wing in the Y-axis direction is caused by torsional motion after being loaded. Under specific pressure loads, the wing did not experience stress concentration or fatigue failure, indicating that the wing has good structural stability. As shown in Figure 14e, the development of the maximum equivalent stress (von Mises) is located at the root of the trailing edge of the wing because there are fixed supports on the connecting rods in this area. The surface of a wing subjected to aerodynamic loads is prone to fatigue or dynamic stress, and it is the first component of the wing to withstand tension under bending loads [40]. Therefore, the stress distribution on the wing after being loaded cannot be ignored in evaluating its structural performance.

### 3.3. Aerodynamic Performance of the Deformable Wing MAV

Quantifying the instantaneous deformation on the wing can help determine the direction and magnitude of fluid dynamics generated by inertial and aerodynamic loads [41]. Therefore, it is crucial to simulate the dynamic changes in the airflow around the wing in order to analyze the aerodynamic mechanism of a deformed wing MAV during flight. Based on the literature [39,42,43,44], the size of the computational domain for the deformable wing MAV model is determined. The calculated average chord length of the wing is 0.1 m. Here, the dimensions of the determined computational domain are 0.4 m in length (approximately four times the average chord length), 0.5 m in width (approximately five times the average chord length), and 0.3 m in height (approximately three times the average chord length). ANSYS ICEM-CFD software is used for meshing geometric models.

Based on the method described in reference [45], five types of computational domain grids are defined, including a coarse mesh model, medium mesh model, fine mesh model, refined mesh model, and re-refinement mesh model. After preliminary calculation comparison, the computational domain with fine mesh is selected as the real CFD solution analysis. The grid structure of the deformation wing MAV computational domain is shown in Figure 15, with a total of approximately 437 million grid elements. A fully validated internal CFD solver was used to simulate incompressible flow through a 3D moving body. The incompressible flow is governed by the Navier–Stokes equation [46] in Formula (8). The k–ε turbulence model is commonly used in CFD simulations of turbulence. The velocity inlet of the computational domain is set with velocity inlet boundary conditions, with an incoming flow velocity of 5 m/s and a specific wing angle of attack (α = 5°) [32]. The outlet of the computational domain is set as the pressure outlet, and its relative pressure is set to 0. The other walls are set as non-slip boundaries. When compiling the mobile grid, the speed of the propeller is set to 5000 rpm [47], and its surface is set as an anti-slip wall. The spatial discretization adopts a second-order upwind scheme, and a pressure-based solver is used for simulation. The pressure velocity coupling scheme employed the SIMPLE solver method. The time step for transient calculation is 0.096 s. The flow field calculation is carried out using a commercial ANSYS Fluent software package based on dynamic grid mode as the flow solver.(8)$$\nabla \cdot u=0;\;\;\;\frac{\partial u}{\partial t}+u\cdot \nabla u=-\frac{1}{\rho }\nabla p+v\nabla^{2}u,$$
where **u** is the velocity vector, *t* is time, and *p* is the air pressure. The fluid density and kinematic viscosity are denoted by *ρ* and *v*, respectively.

Among the multiple rotation cycles of the propeller, one rotation cycle is selected for analysis. Tecplot 360 EX 2023 is used for post-processing. The vortex formations at *t*/*T* = 0.25, 0.5, 0.75, and 0.95 are shown in Figure 16, in which the vortex structures are identified by the iso-surfaces of the Q-criterion (Q = 20). The leading edge vortex (LEV), trailing edge vortex (TEV), and wingtip vortex (TV) continuously form a ring structure on the wing. The vortex passing through the trailing edge forms a circular vortex wake on the tail wing. The leading edge vortex and wingtip vortex are the main aerodynamic effects that generate lift and thrust in deformable wing MAVs. The wingtip vortex moves from the outer region to the center after generation, and during dissipation, the wake radius gradually increases and expands outward. At 0.5 T, the interference between vortices is minimal. However, at 0.75 T, vortices begin to merge and form a mixture of large-scale annular vortices and smaller debris vortices in the wake region. At 0.95 T, vortices gradually form and accumulate on the wing surface at the leading and trailing edges, with an increase in high-pressure areas on the upstream surface and low-pressure positions on the downstream surface. The rapid expansion of this fluid on the wing reflects the increased complexity of the flow field driven by the interaction between the induced flow caused by propeller rotation and the incoming flow [47].

The aerodynamic force acting on the wing can be calculated by integrating the pressure and shear stress on the dorsal and ventral surfaces of the deformed wing MAV model. Then, by converting the total force into horizontal and vertical directions, it is easy to calculate thrust and lift [45]. The dimensionless lift coefficient (*C_L_*) and drag coefficient (*C_D_*) used for aerodynamic characteristics are defined in Formula (9). The three types of vortices formed by the fluid attached to the wing keep the pressure changes on the wing surface within a specific range, which enables the wing to produce reasonable aerodynamic characteristics during flight. The obtained lift, drag, and thrust curves are shown in Figure 17. After calculation, the average lift value generated by the wing over many cycles is 2.7 N. In other words, the air around the biomimetic deformable wing MAV will generate three strong types of vortices, producing enough lift to overcome its own gravity. The CFD simulation results provide important references for the design of lightweight twin-wing coupling for MAVs.(9)$$\left\{\begin{array}{l} C_{L}=\frac{F_{L}}{0.5\rho U^{2}S_{W}}\\ C_{D}=\frac{F_{D}}{0.5\rho U^{2}S_{W}} \end{array}\right.$$
where *F_L_* and *F_D_* represent the lift and drag, respectively. *S_W_* is the area of the wing, which is 74,553.8017 mm^2^.

### 3.4. Aerodynamic Prediction of Biomimetic Deformable Wing MAV

The prediction method of aerodynamic performance based on artificial neural networks (ANNs) can reduce the complexity of CFD analysis. It has been proven feasible to study the aerodynamic characteristics of MAVs and implement more effective parameter optimization methods [48,49,50]. The average thrust obtained based on the CFD simulation mentioned above is 0.3 N. Here, the wavelet neural network is used to predict the subsequent operating conditions based on the calculation results within 0–0.5 T. A three-layer wavelet neural network with multiple inputs, one output, and multiple activation functions is shown in Figure 18 [51]. The output y can be calculated by the following equation:(10)$$y(x)=\sum_{i=1}^{n}w_{i}\varphi_{i}(x)+\sum_{j=1}^{R}v_{j}x_{j}$$
where $\varphi_{i}(x)$ is the activation function for each hidden layer node; $w_{i}$ is the connection weight between the product layer and the hidden layer; and $v_{j}$ is the connection weight between the product layer and the input layer.

Based on thrust data of 0–0.5 T, a wavelet neural network is established to simulate and predict thrust, and the training and testing sets are divided in a 7:3 ratio. Firstly, the offline neural network fitting simulation was validated. The neural network is set to three layers, with 11 neurons in the hidden layer, a learning rate of 0.1, and an upper limit of 1000 training iterations. After network training, 100 sets of data were used as the validation set for testing the model fitting results. The comparison of training set prediction results is shown in Figure 19 (RMSE = 0.0067924). This indicates that the difference between predicted and true values of the model is small, so the model has a good fitting effect and high prediction accuracy. The comparison of the predicted results of the test set is shown in Figure 20 (RMSE = 0.0069812), which indicates that the model has a good fitting effect and high accuracy.

To quantitatively evaluate the simulation effect of the wavelet neural network model on thrust, as shown in Table 3, the correlation coefficient (R^2^), mean absolute error (MAE), mean relative error (MBE), and root mean square error (RMSE) of the training and testing sets were compared and analyzed. It can be seen that the R^2^ of the training set and the test set are close to 0.6, indicating a strong positive correlation between the predicted values and the true values. The fitting effect of the model is good, with MAE, MBE, and RMSE all close to 0, and the maximum not exceeding 0.001. Overall, the model fitting error is small and can be applied to the prediction of actual thrust, providing a certain reference for the trend changes in thrust data. The thrust prediction for the next time step is shown in Figure 21, and it can be seen that the trend in the simulation results is similar to that of the prediction results, which once again demonstrates the feasibility of using neural networks to predict the aerodynamic performance of deformed wings.

For comparison purposes, it is necessary to analyze the lift data obtained from CFD simulation results and experimental tests. This requires the production of a physical prototype and the use of a motion capture system consisting of a high-speed camera system and sensor system. First, a physical prototype is made. Furthermore, 4 mm thick foam plates are cut according to the sector angle of each panel in the 3D model. Specially customized hinges are used to connect each thick panel, with the height of the hinges ensuring that one thick panel is folded between the other two rear panels. The carbon fiber rod is fixed in the middle of the wing with glue, and the tail wing and propeller are both installed on circular carbon fiber rods. The battery and control board are fixed to the thick plate in the middle of the wing with black tape. For easier folding and greater flexibility, small magnets are fixed to the edges of each thick panel with glue, and a thin elastic silicone film is glued onto the wing. The weight of the entire physical prototype after assembly is 24.24 g.

The motion capture system used consists of a high-speed camera, a motion plane of a 3D force sensor, a data processing system, signal transmission equipment, and a support frame for a fixed camera. The motion plane of the object has a length of 0.7 m and a width of 0.55 m. A measurement platform with multiple sets of three-dimensional force sensors in the middle position of the motion plane, with a length of 0.3 m and a width of 0.1 m, is set up. The dimensions of the camera fixed frame are 0.9 m in length, 0.82 m in width, and 0.4 m in height [52]. A cylindrical carbon rod with a diameter of 0.04 m is fixed on the platform of the sensor, and a cyanoacrylate adhesive is used to fix the physical prototype (α = 5°) on the cylindrical carbon fiber rod. The blowing speed of a small fan is assumed to be the incoming flow speed. Two orthogonal synchronized high-speed cameras (Olympus, i-SPEED 3, Tokyo, Japan) installed on the force measurement platform of the motion capture system are used to capture force generation data at a speed of 5000 frames per second. The camera is synchronously triggered for video recording, and the shutter speed is manually set to 50 μs. Due to the large storage space required for long-term video recording, the camera is paired with a computer to record at 1280 × 1024 pixels [53]. To improve the quality of the shooting background, a 300 W projection lamp is used to illuminate the experimental area. As shown in Figure 22, the CFD simulation results are consistent with the prototype test results to a certain extent. The main difference is that the current testing platform is not in a real wind tunnel test environment, but the degree of curve agreement indicates the feasibility of this design, which can be widely promoted and studied as a new type of deformable wing MAV.

## 4. Conclusions

This paper aims to mimic the secondary unfolding performance of the hind wings of the *Protaetia brevitarsis* beetle and design a biomimetic deformable wing MAV with a large folding ratio. The folding performance, structural response characteristics, and aerodynamic mechanism of expandable wings were simulated and analyzed. Firstly, experimental treatment was conducted on the captured beetle’s hind wings, and the mechanism of changes in the folding movement of the hind wings was observed. The origami model was mathematically calculated to obtain the optimal folding angle (sector angle). Then, the four-panel origami model was extended to the six-panel model (double four-panel model), and its folding performance and modal characteristics were analyzed to verify the feasibility of the thick-plate deployable mechanism. Furthermore, the double four-plate mechanism was extended along both the transverse and longitudinal directions, and the spatially deployable deformable wing MAV was constructed. Its kinematics and structural modes were further analyzed. The results indicate that the expandable wing mechanism can fold stably without interference between thick panels. The vibration modes of the wing exhibit bending and torsional deformation, and the deformation of the wing under load is reasonable. Finally, the simulation results of CFD show that the biomimetic deformable wing can gather dense leading edge vortices, wingtip vortices, and trailing edge vortices, thereby generating an average lift (2.7 N) that can overcome the weight of the prototype. In response to the uncertainty and complexity issues in CFD simulation, a neural network is introduced to predict the thrust characteristics under the next operating condition. The comparison with CFD simulation results shows the rationality of the network’s prediction performance. The physical prototype was fabricated and tested to demonstrate the feasibility of biomimetic design by comparing force generation with CFD simulation results.

## Figures and Tables

**Figure 1 biomimetics-10-00320-f001:**
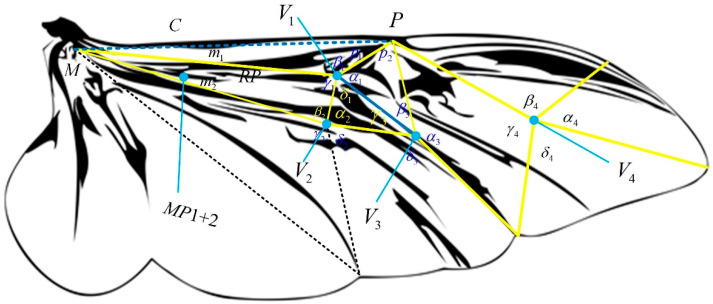
The triangular folding pattern with four angles and a single vertex on the hind wings of the *Protaetia brevitarsis*.

**Figure 2 biomimetics-10-00320-f002:**
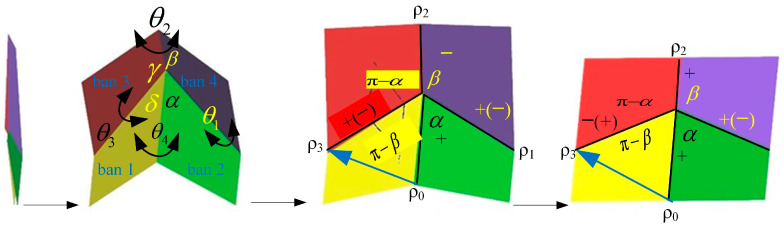
The unfolding process of the cardboard of the equivalent *Protaetia brevitarsis* beetle’s hind wings.

**Figure 3 biomimetics-10-00320-f003:**
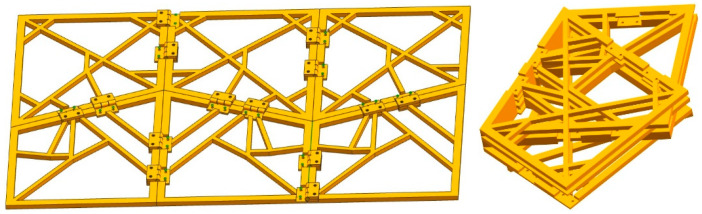
Unfolded and folded configurations of a double four-panel folding model imitating the folding mechanism of hind wings.

**Figure 4 biomimetics-10-00320-f004:**
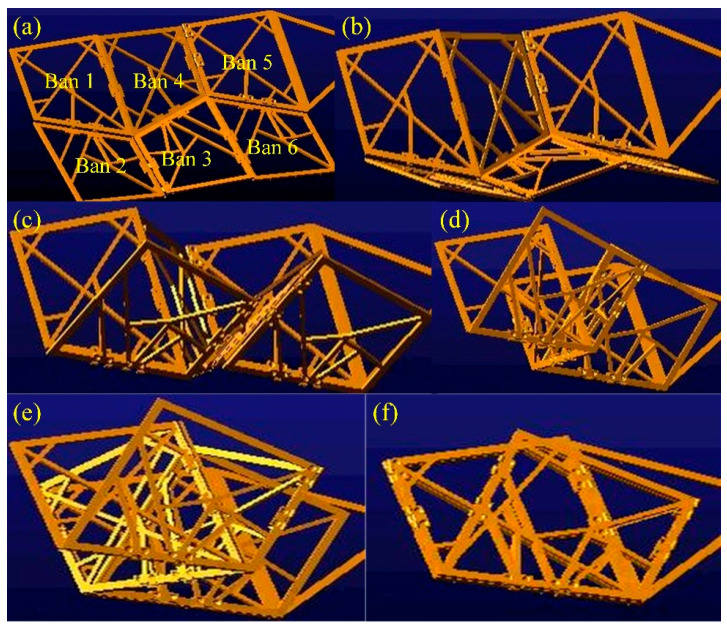
Folding process of biomimetic space deployable double four-panel mechanism. (**a**) Represents the fully unfolded configuration of the deployable mechanism; (**b**) Shows the folding motion at 0.5 s; (**c**) Shows the folding motion at 1.5 s; (**d**) Shows the folding motion at 2.3 s; (**e**) Shows the folding motion at 3.2 s; (**f**) Represents the fully folded configuration of the deployable mechanism.

**Figure 5 biomimetics-10-00320-f005:**
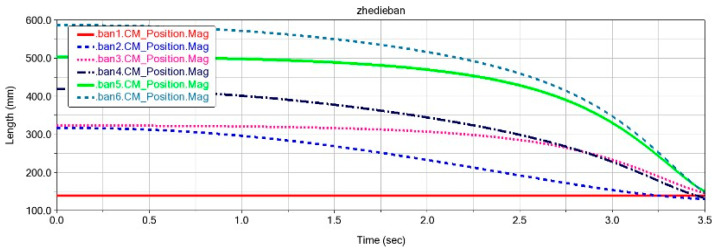
Displacement changes in each cardboard during folding motion of the biomimetic space deployable double four-panel mechanism.

**Figure 6 biomimetics-10-00320-f006:**
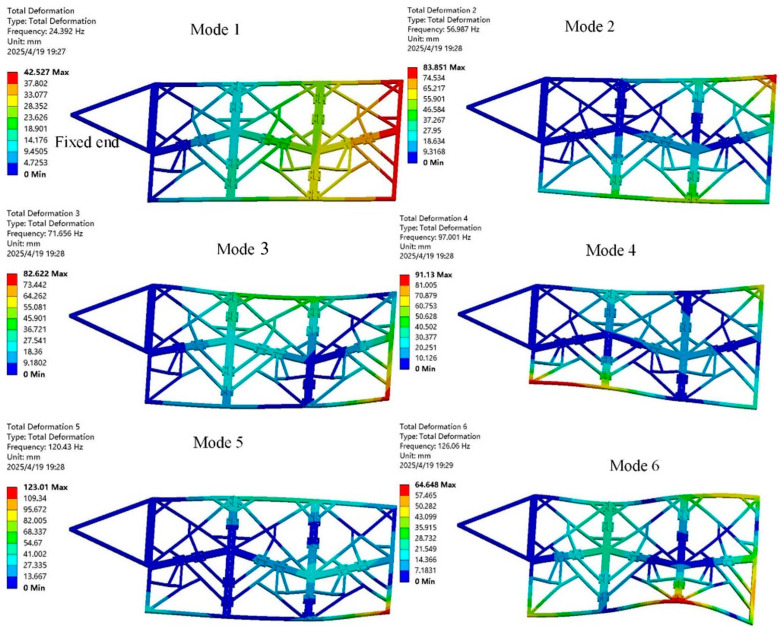
The first six modal shapes of the six-panel deployable mechanism in the unfolded state.

**Figure 7 biomimetics-10-00320-f007:**
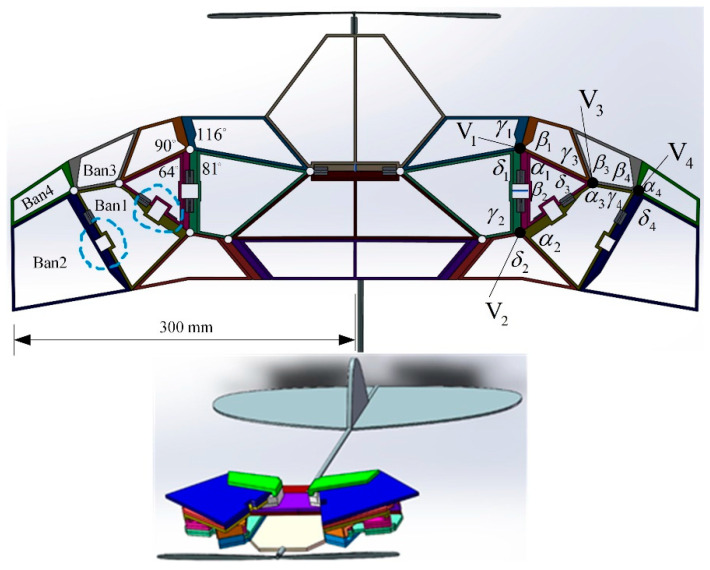
Deformable wing MAV with multi-plate deployable mechanism.

**Figure 8 biomimetics-10-00320-f008:**
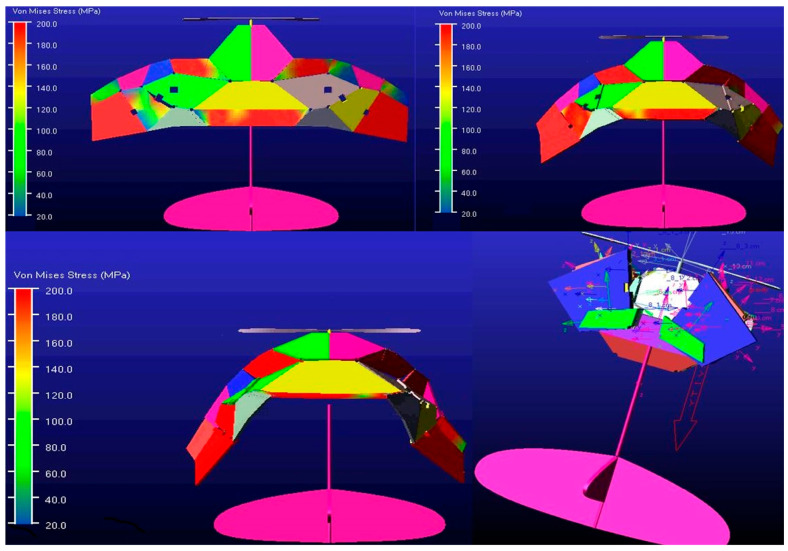
Folding motion of biomimetic space-variable wing MAV.

**Figure 9 biomimetics-10-00320-f009:**
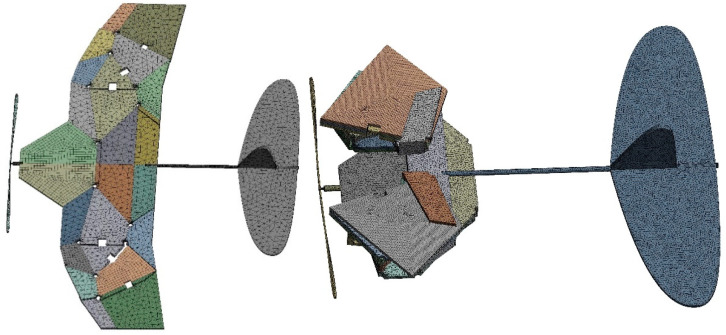
Mesh division of biomimetic deformable wing MAV.

**Figure 10 biomimetics-10-00320-f010:**
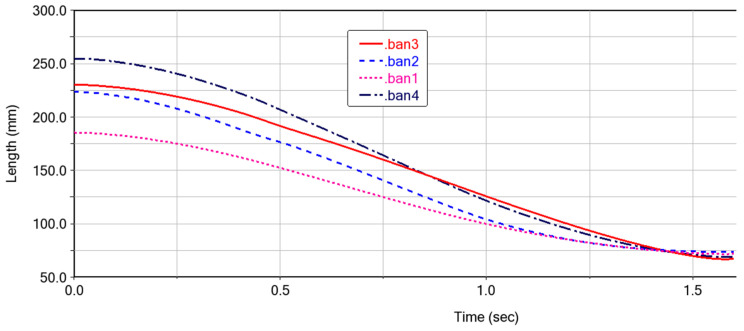
Changes in the motion position of each cardboard during ADAMS simulation.

**Figure 11 biomimetics-10-00320-f011:**
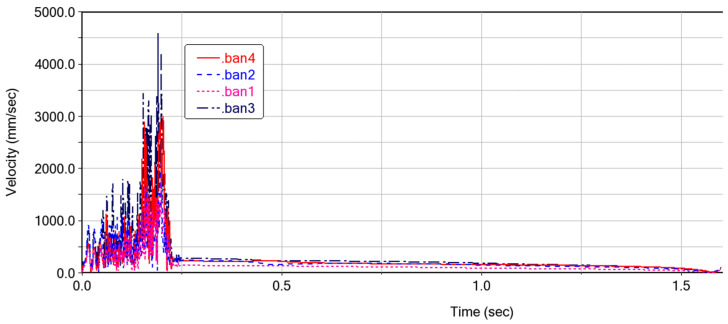
Velocity variation during folding motion in ADAMS simulation.

**Figure 12 biomimetics-10-00320-f012:**
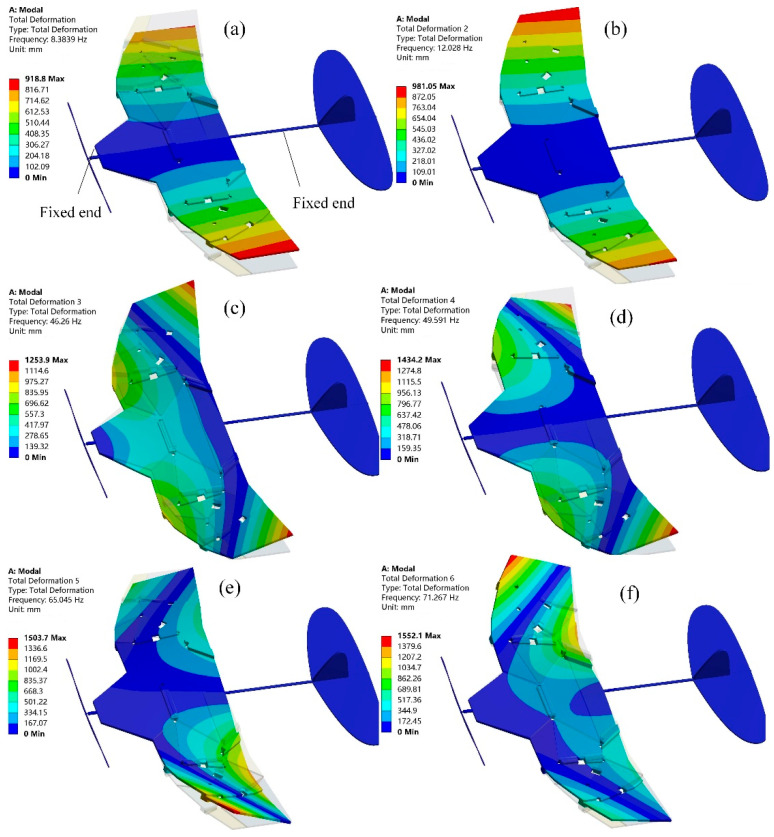
Modal analysis of the deformable wing MAV in unfolded configuration. (**a**) The first mode of vibration; (**b**) The second mode of vibrations; (**c**) The third modal shape; (**d**) The fourth mode of vibration; (**e**) The fifth mode of vibrations; (**f**) The sixth modes of vibrations.

**Figure 13 biomimetics-10-00320-f013:**
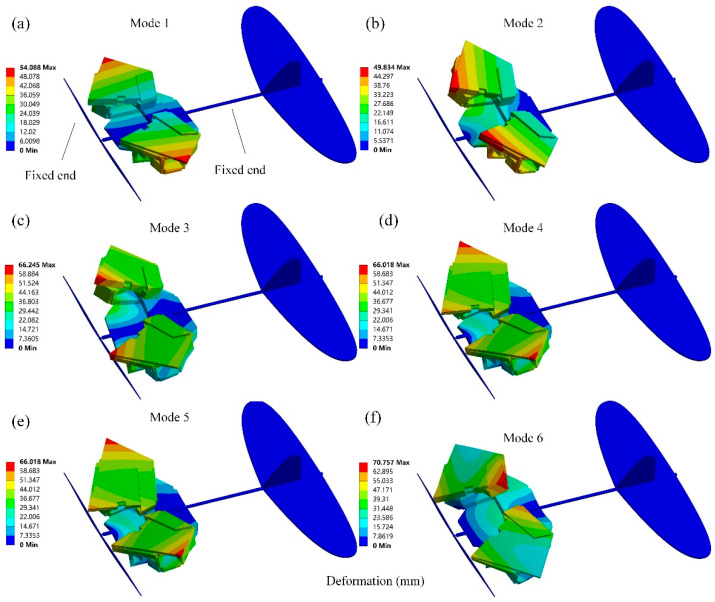
Modal analysis of the deformable wing MAV in deployed configuration. (**a**) The first mode of vibration; (**b**) The second mode of vibrations; (**c**) The third modal shape; (**d**) The fourth mode of vibration; (**e**) The fifth mode of vibrations; (**f**) The sixth modes of vibrations.

**Figure 14 biomimetics-10-00320-f014:**
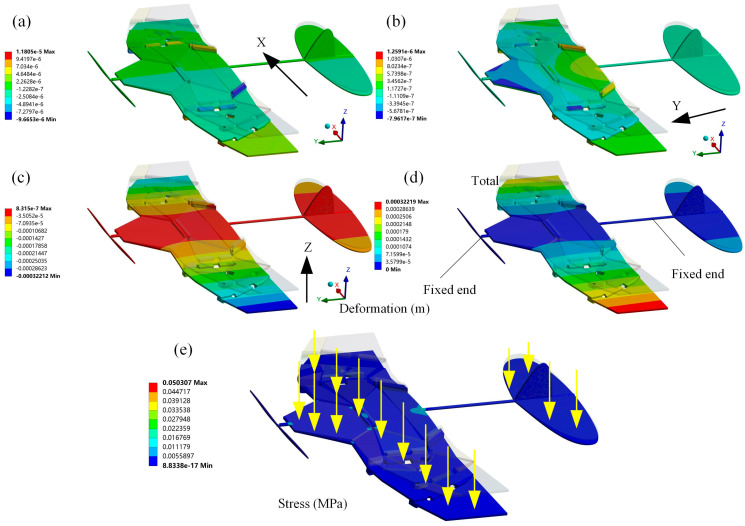
Static analysis of the biomimetic deformable wing MAV. (**a**) Deformation along the x-axis direction; (**b**) Deformation along the y-axis direction; (**c**) Deformation along the z-axis direction; (**d**) The total deformation under pressure load; (**e**) The equivalent (von Mises) stress under pressure load.

**Figure 15 biomimetics-10-00320-f015:**
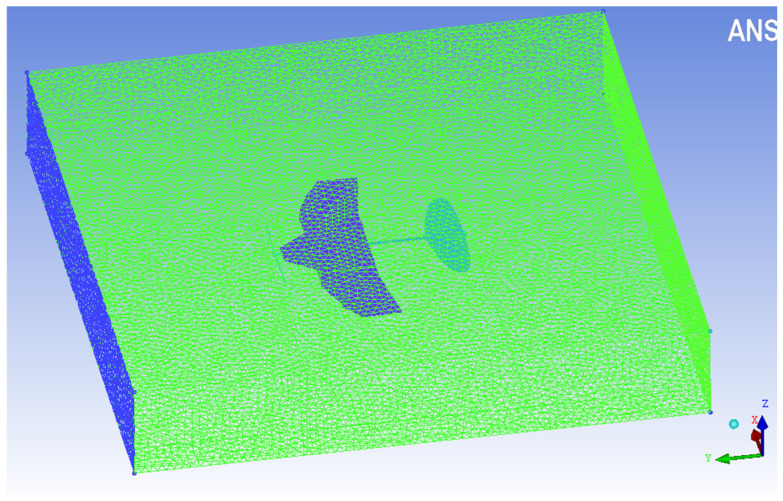
Computational domain mesh of the deformable wing MAV.

**Figure 16 biomimetics-10-00320-f016:**
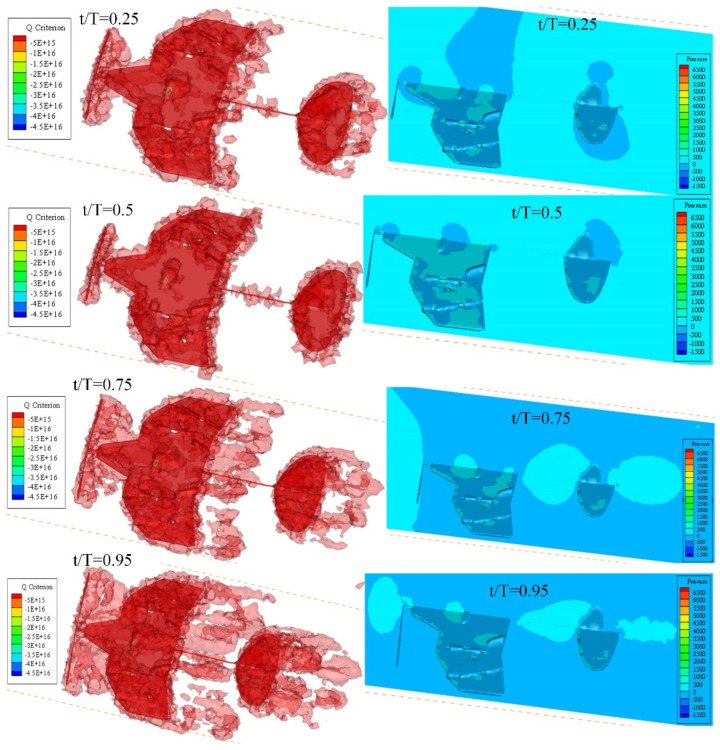
Isometric views of the Q-criterion (Q = 20) and pressure distribution on the wing’s surface at some typical time steps.

**Figure 17 biomimetics-10-00320-f017:**
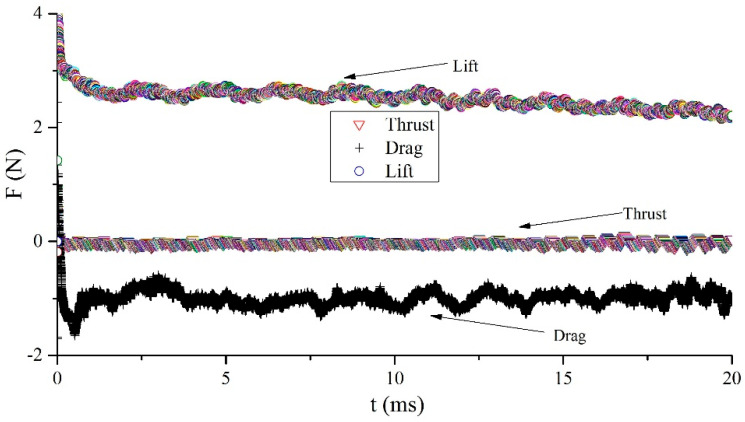
Aerodynamic characteristics of the deformable wing MAV with a specific angle of attack (α = 5°).

**Figure 18 biomimetics-10-00320-f018:**
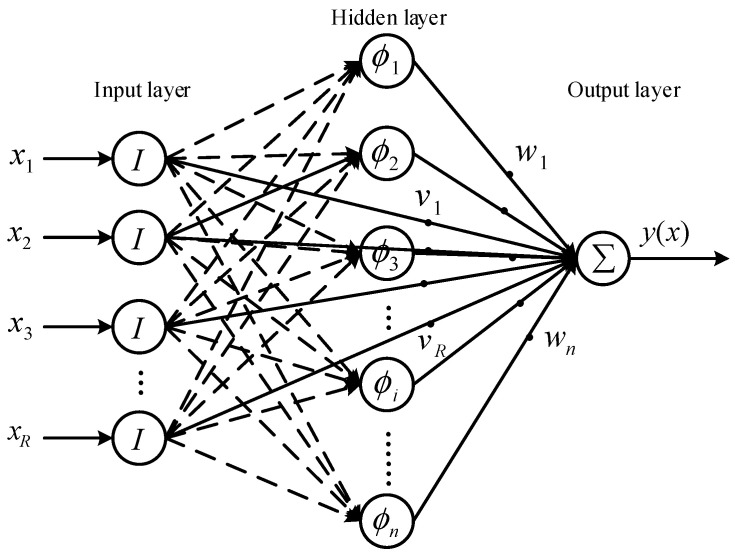
Wavelet artificial neural network.

**Figure 19 biomimetics-10-00320-f019:**
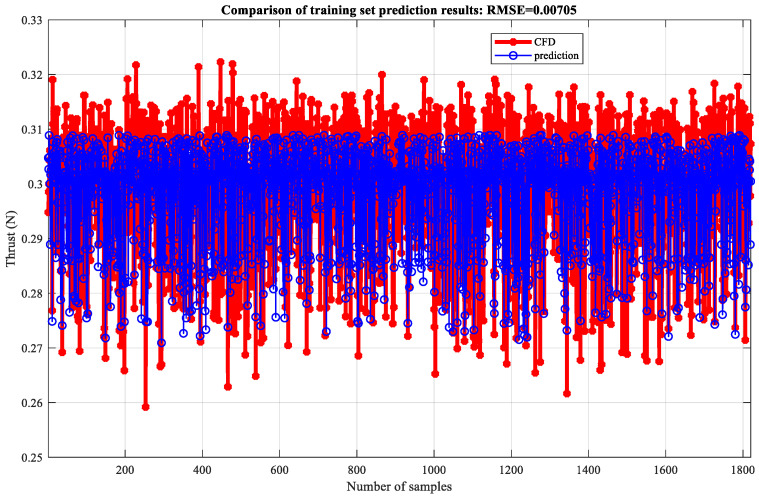
Neural network fitting results of the training set.

**Figure 20 biomimetics-10-00320-f020:**
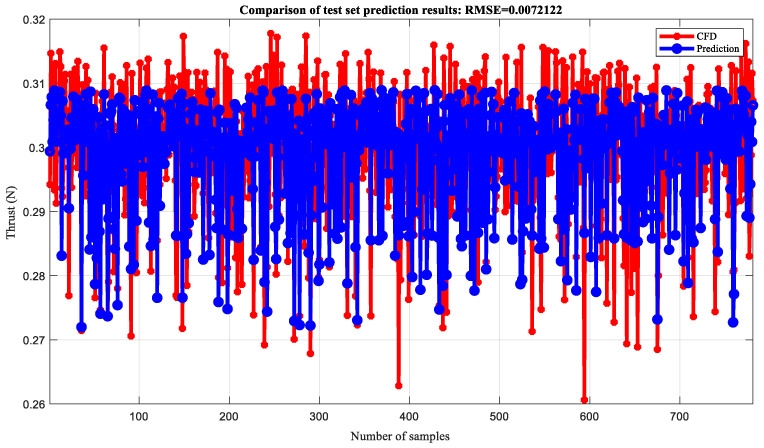
Neural network fitting results of the test set.

**Figure 21 biomimetics-10-00320-f021:**
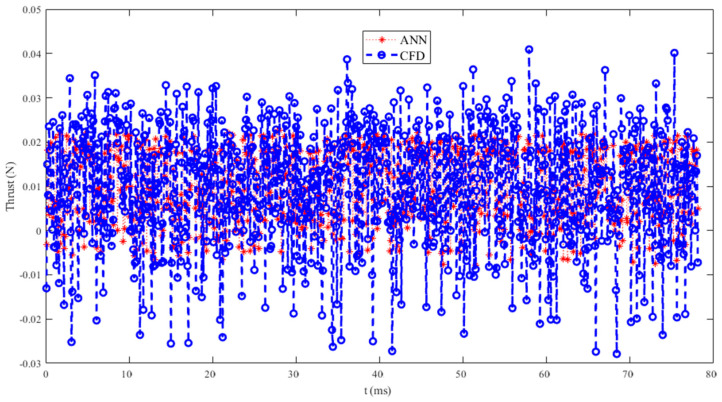
Aerodynamic thrust prediction for the next time step of the deformable wing MAV.

**Figure 22 biomimetics-10-00320-f022:**
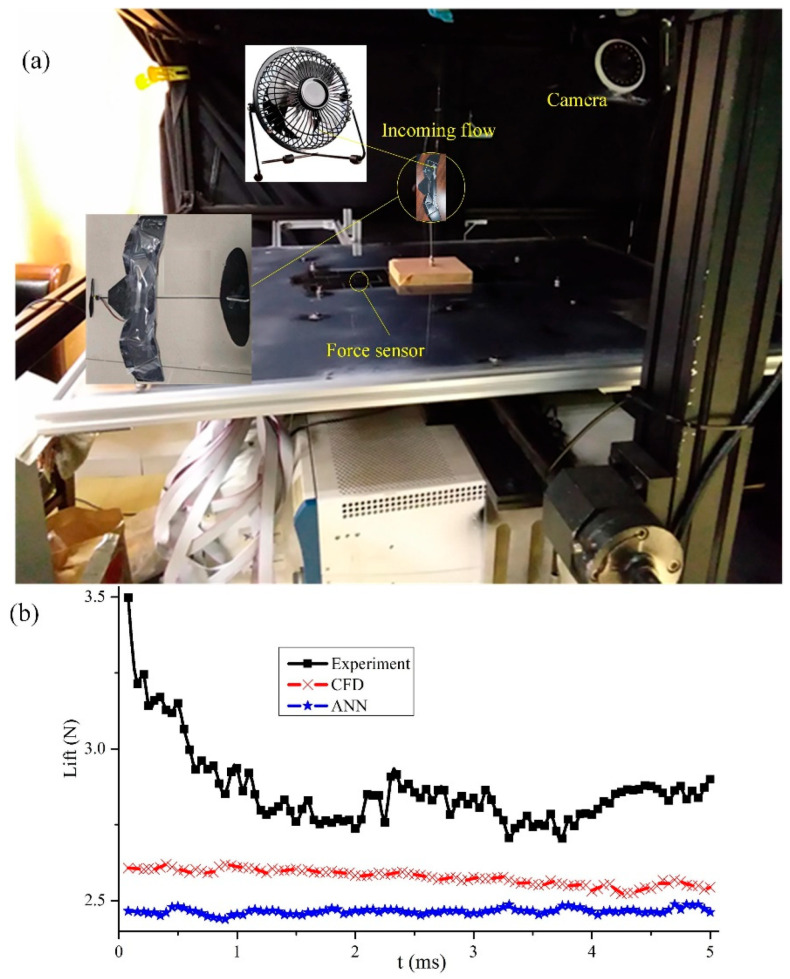
Lift characteristics of the deformed wing MAV (α = 5°) obtained from experiment, numerical simulation, and network prediction. (**a**) The motion capture system is used to measure lift characteristics; (**b**) Comparison of measurement results of motion capture system, calculation results of ANSYS Fluent, and prediction results of artificial neural network.

**Table 1 biomimetics-10-00320-t001:** The first six modes of vibration of the deformable wing MAV in unfolded configuration.

Order	1	2	3	4	5	6
Frequency (H_Z_)	8.3839	12.0280	46.2600	49.5910	65.0450	71.2670

**Table 2 biomimetics-10-00320-t002:** The first six natural frequencies of the deformable wing MAV in folded configuration.

Order	1	2	3	4	5	6
Frequency (H_Z_)	237.24	559.89	959.84	1384.10	1832.60	2038.40

**Table 3 biomimetics-10-00320-t003:** Evaluation effect of thrust simulation based on neural network.

	NN	R^2^	MAE	MBE	RMSE
Thrust	Training set	0.6025	0.0055885	−2.2751 × 10^−5^	0.0067924
	Test set	0.58188	0.005785	8.2895 × 10^−5^	0.0069812

## Data Availability

The data presented in this study are available from the corresponding author upon request. The data are not publicly available due to their relation to another ongoing research project.

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
