# Peer review of "Design and Simulation of a Bio-Inspired Deployable Mechanism Achieved by Mimicking the Folding Pattern of Beetles’ Hind Wings"

_biomimetics, 2025, doi:10.3390/biomimetics10050320_

Round 1
Reviewer 1 Report (Previous Reviewer 3)
Comments and Suggestions for Authors
The manuscript has been in depth modified to yield a good presentation of the ideas and the results. As such, the present form of the article is suitable for publication.
Reviewer 2 Report (Previous Reviewer 2)
Comments and Suggestions for Authors
I would like to thank the authors for addressing my points. The revised manuscript is improved compare to the last submission.
This manuscript is a resubmission of an earlier submission. The following is a list of the peer review reports and author responses from that submission.
Round 1
Reviewer 1 Report
Comments and Suggestions for Authors
This work may be a ‘report’ rather than a paper because the content just includes the summary of data for specific model without physical interpretations including verification. Further, the motivation is not clear, and all content should be reorganized to be a paper with the enhanced title. Primarily, more systematic precise presentation is necessary with the support of native English speaker because too many typo errors are found through the manuscript.
Brief comments:
-Title : Typo error : ...Mimcking...,
. Is ....Simulation Analysis.... correct ?
. Mechanism is duplicated in the title
: Authors should clearly state in a more enhanced form of title.
- Other errors : Ex) a) Subsections 3.3 are duplicated. b) List of some references include et al.
: All other portion should be carefully enhanced without errors.
-Abstract : Why not state the full words instead of D.H parameter ?
-Titles of Figures and Tables should be ‘uniquely stated’ with specific characteristics without overlap the others.
- Section 2 should be renamed for suitable expression.
-Sections 2 and 3 should be smoothly connected for this work.
- For readers’ conveniences, precisely state precisely all subsections of 2 and 3 just before 2.1 and 3.1, respectively.
-Delete line 120 because it is not a report.
-If the Eqns(1) to (7) are not authors’ own, add the references.
-In the figures,
.Why not check the validity of the present results ? Why not compare the previous results with the present study ?
.Use the dimensionless data for more general conclusion.
- Some Figures and data are similar to the direct copy of computer output.
-Conclusion
.Contents are not just a summary of numerical data without verification.
All contents should be rechecked with careful consideration, and more precise presentation is necessary without too detail contents.
Comments on the Quality of English LanguageRefer to the comments.
Author Response
Replies to reviewer 1
- Title : Typo error : ...Mimcking...,
Answer: It has been modified.
- Is ....Simulation Analysis.... correct ?
Answer: It has been modified.
- Mechanism is duplicated in the title
Answer: It has been modified.
- Authors should clearly state in a more enhanced form of title.
Answer: The title of the manuscript has been revised.
- Other errors : Ex) a) Subsections 3.3 are duplicated. b) List of some references include et al.
Answer: It has been modified.
- All other portion should be carefully enhanced without errors.
Answer: The manuscript has been rewritten.
- Abstract: Why not state the full words instead of D.H parameter ?
Answer: The abstract content has been modified.
- Titles of Figures and Tables should be ‘uniquely stated’ with specific characteristics without overlap the others.
Answer: The graphics and tables have been modified.
- Section 2 should be renamed for suitable expression.
Answer: It has been modified.
- Sections 2 and 3 should be smoothly connected for this work.
Answer: The content of these two parts has been revised and elaborated in the paragraphs.
- For readers’ conveniences, precisely state precisely all subsections of 2 and 3 just before 2.1 and 3.1, respectively.
Answer: The content of these paragraphs has already been rewritten.
- Delete line 120 because it is not a report.
Answer: It has been modified.
- If the Eqns(1) to (7) are not authors’ own, add the references. In the figures,
Answer: References have been added.
- Why not check the validity of the present results? Why not compare the previous results with the present study?
Answer: Comparative analysis graphics were added to the manuscript.
- Use the dimensionless data for more general conclusion.
Answer: The result content has been restated.
- Some Figures and data are similar to the direct copy of computer output.
Answer: Graphics and data are obtained through post-processing software.
- Contents are not just a summary of numerical data without verification.
Answer: Added cited references in the corresponding paragraph statements.
- All contents should be rechecked with careful consideration, and more precise presentation is necessary without too detail contents.
Answer: All the content of the manuscript has been rewritten.
Thank you very much. With best wishes,
Reviewer 2 Report
Comments and Suggestions for Authors
The authors of the entitled manuscript “Design and Simulation Analysis of a Spatially Deployable Mechanism Mimcking the Folding Mechanism of Beetles' Hind Wings”
- The current abstract is missing to briefly describe the results and the novel work for this study.
- The introduction section should show the current CFD methodologies used for this study. The authors based their findings on the state of the art and show the current gap in the research area.
- Figure 8 data is not clear and difficult to read.
- Line 249 mentioned the ANSYS workbench, but which software under Ansys is used to create the mesh shown in Figure 9?
- The results presented in Figure 12 show the computational domain which helped you to find the total deformation.
- I could not see information to assess the mesh convergence.
- Figures 15 and 16 are not clear. Also, why have the authors not used CFX or Fluent for the CFD analysis since they are working on the ANSYS workbench? Again, the computational domain for the CFD and the type of analysis used there should be shown.
- Is this study representative of one-way fluid-structure interaction or not even considered?
Author Response
Replies to reviewer 2
- The current abstract is missing to briefly describe the results and the novel work for this study.
Answer: The abstract content has been modified.
- The introduction section should show the current CFD methodologies used for this study. The authors based their findings on the state of the art and show the current gap in the research area.
Answer: The relevant paragraphs in the introduction have been revised.
- Figure 8 data is not clear and difficult to read.
Answer: The graphic has been modified.
- Line 249 mentioned the ANSYS workbench, but which software under Ansys is used to create the mesh shown in Figure 9?
Answer: ANSYS workbench was used to create the mesh.
- The results presented in Figure 12 show the computational domain which helped you to find the total deformation.
Answer: Modal analysis can help understand the corresponding characteristics of the mechanism.
- I could not see information to assess the mesh convergence.
Answer: The convergence data of the grid is described in the upper paragraph of Figure 9.
- Figures 15 and 16 are not clear. Also, why have the authors not used CFX or Fluent for the CFD analysis since they are working on the ANSYS workbench? Again, the computational domain for the CFD and the type of analysis used there should be shown.
Answer: These contents have been modified. ANSYS ICEM-CFD software is used for meshing. ANSYS Fluent is used for solving calculations. Tecplot 360 EX 2023 is used for post-processing.
- Is this study representative of one-way fluid-structure interaction or not even considered?
Answer: Unidirectional fluid structure coupling is set up in Fluent software.
Thank you very much. With best wishes,
Reviewer 3 Report
Comments and Suggestions for Authors
The title of the manuscript is: "Design and simulation analysis of a spatially deployable mechanism mimicking the folding mechanism of beetles' hind wings". The paper is well written, however some issues should be addressed before the article can be published:
- There is a spelling issue in the title: it should be "mimicking" .
- The study does not address the transition from closed to fully deployable state and its vice versa. This is an important issue, and it should be discussed in the paper.
- The weight of the whole mechanism is not mentioned. Also, the propulsion issue is missing. Those things should be addressed in the content of the paper.
Author Response
- There is a spelling issue in the title: it should be "mimicking".
Answer: It has been modified.
- The study does not address the transition from closed to fully deployable state and its vice versa. This is an important issue, and it should be discussed in the paper.
Answer: It has been modified.
- The weight of the whole mechanism is not mentioned. Also, the propulsion issue is missing. Those things should be addressed in the content of the paper.
Answer: It has been modified.
Thank you very much. With best wishes,